# Single-Cell RNA Transcriptome Profiling of Liver Cells of Short-Term Alcoholic Liver Injury in Mice

**DOI:** 10.3390/ijms24054344

**Published:** 2023-02-22

**Authors:** Ligang Cao, Di Wu, Lin Qin, Daopeng Tan, Qingjie Fan, Xiaohuan Jia, Mengting Yang, Tingting Zhou, Chengcheng Feng, Yanliu Lu, Yuqi He

**Affiliations:** 1Guizhou Engineering Research Center of Industrial Key-Technology for Dendrobium Nobile, Zunyi Medical University, Zunyi 563000, China; 2Joint International Research Laboratory of Ethnomedicine of Ministry of Education, Zunyi Medical University, Zunyi 563000, China

**Keywords:** short-term alcoholic liver disease, scRNA-seq, hepatocytes, endothelial cells, Kupffer cells, transcription factor

## Abstract

Alcoholic liver disease (ALD) is currently considered a global healthcare problem with limited pharmacological treatment options. There are abundant cell types in the liver, such as hepatocytes, endothelial cells, Kupffer cells and so on, but little is known about which kind of liver cells play the most important role in the process of ALD. To obtain a cellular resolution of alcoholic liver injury pathogenesis, 51,619 liver single-cell transcriptomes (scRNA-seq) with different alcohol consumption durations were investigated, 12 liver cell types were identified, and the cellular and molecular mechanisms of the alcoholic liver injury were revealed. We found that more aberrantly differential expressed genes (DEGs) were present in hepatocytes, endothelial cells, and Kupffer cells than in other cell types in alcoholic treatment mice. Alcohol promoted the pathological processes of liver injury; the specific mechanisms involved: lipid metabolism, oxidative stress, hypoxia, complementation and anticoagulation, and hepatocyte energy metabolism on hepatocytes; NO production, immune regulation, epithelial and cell migration on endothelial cells; antigen presentation and energy metabolism on Kupffer cells, based on the GO analysis. In addition, our results showed that some transcription factors (TFs) are activated in alcohol-treated mice. In conclusion, our study improves the understanding of liver cell heterogeneity in alcohol-fed mice at the single-cell level. It has potential value for understanding key molecular mechanisms and improving current prevention and treatment strategies for short-term alcoholic liver injury.

## 1. Introduction

Liver is a combination of liver parenchyma cells and non-parenchymal cells (NPCs). Hepatocytes constitute the liver parenchyma and approximately account for 78% of liver volume [1]. NPCs include liver sinusoidal endothelial cells, macrophages, B cells, T cells, hepatic stellate cells, Kupffer cells, and bile duct epithelial cells [2,3]. All liver cells play an important role in maintaining the liver’s physiological homeostasis. When the cell function is impaired or its composition becomes abnormal, this can cause diseases such as fatty liver disease, cirrhosis and hepatocellular carcinoma (HCC).

Alcoholic liver disease (ALD) is a disease caused by the decline and failure of a variety of liver functions due to short-term or long-term alcohol intake, including alcoholic fatty liver (AFL), alcoholic hepatitis (AH), alcoholic cirrhosis and liver cancer [4]. ALD is the leading cause of death with alcohol consumption, and more than 50% of deaths related to liver cirrhosis in the world can be attributed to alcohol [5]. A large number of studies has demonstrated that alcoholic liver injury involves a variety of biological processes, including changes in alcohol metabolic enzymes, liver steatosis, inhibition of AMPK signaling pathway, oxidant/antioxidant imbalance, hepatocyte hypoxia, NF-κB and TLR4 signaling pathway activation, hepatocyte apoptosis, and activation of hepatic stellate cells [6,7,8]. However, due to the liver tissue containing heterogeneous cell mixtures, the pathogenesis of alcoholic liver injury in various liver cell types remains unclear.

Bulk RNA sequencing and proteome of the liver tissues cannot distinguish the gene expression of different cell types and do not provide information about cell-cell interaction and microenvironment composition. Single-cell RNA-sequencing (scRNA-seq) has been developing rapidly and has been applied in many research fields in recent years, such as the construction of cell maps, as well as research on the development process of embryo and liver, and the pathogenesis of diseases [9,10,11,12]. Compared with bulk transcriptome sequencing, scRNA-seq also enables us to identify normal and pathogenic cell populations in the liver [13]. However, the change process of various cells during the progression of alcoholic liver injury is still unclear and there are few scRNA-seq studies on liver development in alcoholic liver injury mice. Therefore, comprehensive studies on cell type-specific composition and function are required to assess the detailed molecular mechanisms behind alcoholic liver injury and provide a comprehensive understanding of disease pathogenesis.

To extend our understanding of subsets of each cell type involved in the development and progression of short-term alcoholic liver injury and cell subsets primarily affected by alcohol, we performed short-term pathological models of alcoholic liver injury. ScRNA-seq was used to analyze mice’s hepatocytes and NPCs at four critical stages of liver development in short-term alcoholic liver injury. Our analysis reveals hepatocyte and NPCs gene expression landscapes in the liver during the pathogenesis of alcoholic liver injury, as well as gene regulation and transformation occurring at the onset. In addition, we identify several cell populations that respond most to alcohol and their aberrantly activated transcription factors during the progression of alcoholic liver injury. This study reveals the heterogeneity, complexity and gene expression changes of liver cells and provides novel insights into the fundamental biology and pathology of alcoholic liver injury.

## 2. Results

### 2.1. Short-Term Alcohol Consumption Induced Liver Injury in Mice

The study confirmed that short-term consumption of alcohol caused liver injury in mice. Excessive alcohol consumption for 1 day could increase serum levels of ALT, AST [14]. Liver steatosis could be induced by excessive alcohol consumption for 3 to 7 days [15]. Continuous excessive alcohol consumption for 14 days could cause liver cell edema and necrosis in mice [16]. To elucidate dynamic changes during alcoholic liver injury pathogenesis, we performed mouse models from four key time points of alcoholic liver injury development. The results showed that the serum ALT was significantly upregulated at AG1, AG3, AG7 and AG14; serum AST was significantly upregulated at AG3 and AG7; serum HDL-C levels were significantly downregulated at AG1, AG3, AG7 and AG14; serum LDL-C levels were significantly downregulated at AG1, AG3 and AG7; serum TC levels of mice were downregulated at AG1 and upregulated at AG3, AG7, and AG14 (Figure 1A). In addition, alcohol exposure significantly increased liver and stomach coefficients in alcohol-fed mice (Appendix A). H&E staining results of liver tissue (Figure 1B) show that distilled water (BG group) has no significant effect on the pathological characteristics of liver tissue. Compared to the BG group, the paraffin sections AG group showed obvious hepatocyte necrosis, edema and nucleus pyknosis. These results suggested that short-term administration of 53% alcohol (10 mL/kg) could cause liver damage in mice.

### 2.2. Single-Cell Expression Atlas of Healthy and Alcoholic Liver Injury Mouse Livers

To elucidate liver cell complexity, heterogeneity and their dynamic changes in the pathogenesis of alcoholic liver injury, scRNA-seq was performed on liver cells from healthy and alcoholic liver injury mice at different times. Each cell had on average 33,918 reads and the exon reads took up 70.94% of the total reads. After removing low-quality cells, 50,274 single-cell transcriptomes were reserved and analyzed, including 10,439 from BG group, 10,164 from AG1 group, 9459 from AG3 group, 9260 from AG7 group, and 10,952 from AG14 group (Appendix A). Subsequently, the expression matrix of each cell was created and analyzed using the Seurat R package. t-SNE plot showed that the liver cells were evenly distributed in each group, no significant intergroup batch effect was observed among the 5 groups (Appendix A). A total of 38 clusters were identified with a resolution of 0.8 (Appendix A). All liver cells could be assigned to 12 major liver cell types based on the expression of marker genemarker gene expression and SingleR package (version 2.0.0) (Figure 2A,B). They were B cells (B, marked with *Ms4a1*), Cycling (marked with *Birc5*), dendritic cells (DCs, marked with *Slglech*), endothelial cells (Endo, marked with *Kdr*), granulocyte (Gran, marked with *S100a9*), hepatocytes (Hep, marked with *Alb*), hepatic stellate cells (HSCs, marked with *Dcn*), monocyte or monocyte-derived macrophages (Mo/MoMF, marked with *Ccr2*), natural killer cells (NK, marked with *Klrb1c*), plasma cell (Plasma, marked with *Jchain*), T cells (T, marked with *Cd3d*) and Kupffer cells (KCs, marked with *Clec4f*). The proportions of cells in each sample are shown in Figure 2D and Appendix A. The top three differentially expressed genes for each identified cell type are listed in Figure 2C; enrichment analysis further confirmed the cell identity (Appendix A).

### 2.3. Persistent Alcohol Stimulation Primarily Affects Gene Expression in Mouse Hepatocytes, Endothelial Cells, and Kupffer Cells

The molecular and biochemical mechanisms of alcoholic liver injury pathogenesis and the exact triggers of disease progression are not completely understood. Many mechanisms have been postulated to be involved in the pathology of alcoholic liver injury, such as mitochondrial damage, oxidative stress, endoplasmic reticulum stress, inflammatory pathway activation and dysfunctional lipid metabolism [4,17,18]. A better understanding of these mechanisms and the role of different cell types in this process is essential for the prevention and treatment of alcoholic liver injury. Therefore, this study has collected 713 genes related to the pathogenesis of ALD and investigated their changes in different liver cell types. These ALD-associated genes were matched by all liver cell types to a scRNA-seq gene expression matrix, and DEGs (*p* < 0.01 vs. BG group) in each cell type were screened. We observed the expression of these DEGs in different cell types of the liver of alcoholic liver injury mice at different time points (Figure 3A). The heatmap showed that only a few genes in each cell type were continuously up-regulated or down-regulated with the prolongation of the alcohol infusion, and these genes may continue to play a role in the process of alcoholic liver injury. The R package cluster was used to screen and show the DEGs that vary continuously in different cells and these DEGs were mainly distributed in Hep, Endo and KCs (Figure 3B, Appendix A). In addition, we also performed a heatmap display of the DEGs (*p* < 0.01 vs. BG) of each cell type that was not collected (Figure 3C). The results were consistent with previous studies. Only some of the DEGs continued to change with the prolongation of alcohol and they were mainly present in Hep, KCs and Endo; there were 591 DEGs in Hep, 596 DEGs in KCs and 217 DEGs in Endo (Figure 3D). These results indicated that sustained alcohol stimulation predominantly affects hepatocyte, endothelial cell, and Kupffer cell gene expression in mouse liver.

### 2.4. Abnormal Regulation of Genes and TFs in Hepatocytes of Alcoholic Liver Injury Samples

Hepatocytes are the predominant cell in the liver, comprising about 60% of liver cells, and play an important role in detoxification, lipid metabolism, protein metabolism and glycogenolysis [19]. In the present study, 7739 hepatocytes from the livers of healthy and alcoholic liver injury mice were analyzed, and the result showed that hepatocyte markers *Alb*, *Apoa1*, *Apoa2* and *Ass1* were enriched (Figure 4A). Hepatocytes were generally less proliferative cells. Most hepatocytes were assigned to the G1 phase, and G1 phase cells increased in AG7 and AG14 compared with BG (Figure 4B). These results indicated that alcohol could inhibit the proliferation of hepatocytes.

Alcohol abuse causes an imbalance in the oxidant/antioxidant status of individuals and reduces their ability to regulate oxidative stress [20]. Alcohol induces *Cyp2e1* to induce oxidative stress, while *Cyp2a5* can be induced to inhibit alcohol-induced oxidative stress [21,22]. In this study, 5 oxidative stress-related genes were mainly expressed in hepatocytes (Appendix A). *Gpx1* were significantly down-regulated, while *Cyp2a5*, *Cyp2e1*, *Mt2* and *Sod* were significantly up-regulated in the hepatocytes of alcohol-fed mice (Figure 4C). This suggested that alcohol reduced the detoxification capacity of hepatocytes, and enhanced oxidative stress. Iron overload in the liver can aggravate liver damage by promoting lipid peroxidation, oxidative stress and iron death [23]. Three iron-related genes were mainly expressed in hepatocytes (Appendix A). The *Trf* and *Tmprss6* were significantly up-regulated in hepatocytes while *Hamp* was significantly down-regulated in all cell types in alcoholic liver injury mice (Figure 3B and Figure 4D). Steatosis is an early manifestation of alcoholic liver injury and may increase the susceptibility of the liver to secondary injury. Lipidomic analysis also showed that alcohol could promote the accumulation of lipids in hepatocytes [8]. Many genes of fatty acid synthesis and metabolism were mainly expressed in hepatocytes in this study (Appendix A). Fatty acid synthesis genes *Scd1*, *Acsl1*, and *Acsl5* were significantly up-regulated in hepatocytes of alcohol-fed mice (Figure 4E). Fatty acid degradation genes *Acad1*, *Acadm*, *Acat1*, *Acat3*, *Eci1*, *Gcdh*, *Hadn*, and *Hadhhb* were significantly down-regulated, but *Cpt1a*, *Cyp4a10*, and *Cypa14* were significantly up-regulated in hepatocytes of alcohol-fed mice (Figure 4F). In addition, some genes of cholesterol metabolism were mainly expressed in hepatocytes (Appendix A). *Acaa2*, *Angptl3*, and *Apoc1* were significantly down-regulated, while *Angptl4*, *Apoa1*, *Apoa4*, *Apob*, *Apoe*, *Apoh*, *Cyp27a1*, *Cyp39a1*, *Cyp4a31*, *Cyp8b1*, *Ehhadh*, and *Lcat* were significantly up-regulated compared with the BG group in hepatocytes (Figure 4G). These results indicated that alcohol could increase fat synthesis, decrease degradation and disorder cholesterol metabolism in hepatocytes, resulting in the accumulation of fat in hepatocytes causing alcoholic fatty liver. The complement system is an important part of the innate immune defense, and activation of complement through classical and alternative pathways was detected in the livers of patients with alcohol-associated hepatitis [24]. In our study, it was found that complement and coagulation genes were mainly expressed in hepatocytes (Appendix A), and *C1s1*, *C3*, *C4b*, *C4bp*, *Cfh*, *Cfi*, *Cfhr2*, *F12*, *F2*, *F5*, *F7*, *Fga*, *Fgb*, *Fgg*, *Hc*, *Kng1*, *Kng2*, *Plg*, *Serpinc1*, *Serpingl* and *Vtm* were significantly upregulated in hepatocytes of alcoholic liver injury mice (Figure 4H). Hence, the complement system and anticoagulant system in hepatocytes might be activated during alcoholic liver injury. In addition, we annotated the GO function of the DEGs that continuously changed with ethanol in hepatocytes and analyzed their biological processes (Figure 4I). The 190 up-regulated DEGs were mainly enriched in the regulation of lipid metabolism, alcohol metabolism, acylglycerol metabolism, triglyceride metabolism and fatty acid biosynthesis. The 401 downregulated DEGs were mainly enriched in aerobic respiration, oxidative phosphorylation, cellular respiration, ATP metabolic, energy generation, respiratory electron transport chain, and mitochondrial ATP synthesis coupled electron transport. These results suggest that short-term alcohol injury to mouse hepatocytes mainly involves lipid metabolism, oxidative stress, iron overload, complement and coagulation, and energy metabolism.

SCENIC analysis was performed to assess changes in transcription factors (TFs) in alcoholic liver injury mice. In this way, we predicted specific TFs in alcoholic liver injury hepatocytes (Figure 4J). SCENIC analysis of hepatocytes revealed that some TFs were significantly activated in hepatocytes from alcohol-fed mice, including *Xbp1*, *Stat3*, *Rxra*, *Nfic*, *Nfia*, *Hif*, *Hnf4a*, *Sf1*, *Nfat5*. However, the activity of *Nr1i2* and *Ppara* decreased after 1-day of alcohol and then increased with continued stimulation by alcohol. Functional enrichment of TFs target genes revealed that *Xbp1* regulates the unfolded protein response (UPR) associated with endoplasmic reticulum stress [25]. *Stat3* and *Rxra* are involved in the acute phase response and coagulation. *Nfic*, *Nfia*, *Nr1i2*, *Hlf*, *Hnf4a*, *Rora*, and *Ppara* are involved in lipid synthesis/metabolism processes (Appendix A). These results indicated that continuous alcohol consumption could significantly activate ER stress, acute phase response proteins, coagulation system and lipid synthesis/metabolism processes in hepatocytes.

### 2.5. Abnormal Gene Expression and TFs of Endothelial Cells in Alcoholic Liver Injury Samples

Liver endothelial cells, including sinusoidal endothelial cells (LSEC), vascular endothelial cells and lymphatic endothelial cells (LyECs), play a key role in liver homeostasis, regulating intrahepatic vascular pressure and immune cell function [26]. Traditional immunofluorescence, flow cytometry, isolation of endothelial cells for RNA-seq and other methods are still limited by antigens and immune reagents [27]. scRNA-seq can provide abundant cell markers and cell function profiles and has been used to reveal the region specificity and function of mouse and human liver endothelial cells [27,28]. We analyzed 8399 endothelial cells in total, which highly express the endothelial cell markers *Kdr*, *Pecam1*, *Lyvel* and *Oit3* (Figure 5A). The cells of all groups were less proliferative, indicating that alcohol did not significantly affect the proliferation of hepatic endothelial cells (Figure 5B). Some evidence supports that LSEC injury is increased when *Nos3* is inhibited [29]; down-regulation of *Klf2* and *Nos3* can reduce NO production and lead to LSEC dysfunction [26]. We found that *Klf2* was down-regulated at AG1 and AG3 and *Nos3* were downregulated at AG14 compared to the BG group (Figure 5C). In addition, we enriched Go functions of DEGs that continuously changed with alcohol in liver endothelial cells and analyzed their biological processes (Figure 5D). The 79 up-regulated DEGs were enriched in blood pressure regulation, active regulation of inflammatory response, regulation of leukocyte differentiation, regulation of T cell activation, gliogenesis, hematopoiesis regulation and other functions. The 138 down-regulated DEGs were enriched in amebic cell migration, epithelial cell migration, epithelial migration, tissue migration, endothelial cell migration and other functions. These results indicated that the effects of alcohol on mouse endothelial cells involve the reduction of NO production, blood pressure regulation, inflammatory reaction, and epithelial cell migration.

SCENIC analysis showed the changes of TFs in endothelial cells (Figure 5E). The activities of TFs *Bcl3* and *Klf6* were significantly enhanced at AG1 and subsequently returned to normal levels, but *Nfe2l1* decreased at AG1 and recovered under continuous alcohol stimulation. TFs target genes enrichment analysis showed that *Bcl3* was enriched in immune cell differentiation and cytokine pathways, while *Klf6* was enriched in glucose synthesis or metabolism (Appendix A).

### 2.6. Abnormal Gene Function and TFs in Kupffer Cells of Alcoholic Liver Injury Samples

Kupffer cells are resident macrophages in the liver and can participate in the development of alcoholic liver injury by activating cytokines and chemokines [30]. Because the Kupffer cell is difficult to isolate from the human liver and has a complex developmental process, less is known about it. We detected 344 Kupffer cells, which highly expressed the Kupffer cells markers *Clec4f*, *Timd4* and *Vsig4* (Figure 6A). Cell cycle analysis showed that the proportion of G2M cells in Kupffer cells decreased at AG1, AG7 and AG14 (Figure 6B). In addition, the GO enrichment of persistently changing DEGs in Kupffer cells of the liver of alcoholic liver injury mice showed that 304 up-regulated genes were enriched in intracellular receptor signaling pathways, regulation of mRNA metabolic processes, mRNA processing, RNA splicing and translation regulation, while 292 down-regulated genes were enriched in the processes of ATP metabolism, antigen processing and presentation of exogenous antigens, aerobic respiration, cellular respiration, antigen processing and presentation (Figure 6C). The above results indicate that the effect of alcohol on Kupffer cells in mice involves processes such as antigen presentation and cellular energy metabolism.

SCENIC analyzed the changes of TFs activity in Kupffer cells (Figure 6D). The activities of TFs *Spi1*, *Spic* and *Elf4* on AG1 increased and decreased with continuous alcohol stimulation. The activities of *Zmiz1*, *Z1b1*, *Gata4* and *Sox18* decreased in AG and increased with the stimulation of alcohol. Functional enrichment of TFs target genes showed that *Spic*, *Elf4*, *Sox18*, *Spi1* were involved in immune function, and *Zeb1* and *Gata4* were enriched in relation to cell migration (Appendix A).

## 3. Discussion

Liver cells are mainly divided into hepatocytes and NPCs, and they maintain the microenvironment to keep homeostasis or break the balance under a pathologic environment [31]. The occurrence of ALD involves liver steatosis, hepatocyte necrosis and apoptosis, oxidative stress, immunity and inflammation [32]. Studying the changes in these processes in different cell types is necessary for the treatment and prevention of alcoholic liver injury. Liver transcriptome research based on scRNA-seq can obtain information on different liver cell types, which is conducive to understanding the changes of different liver cells. Therefore, we performed large-scale unbiased scRNA-seq to accurately and systematically profile mice livers with healthy and alcohol-induced liver injury. The large-scale dataset and deep analysis of scRNA-seq truly recognize the heterogeneity and complexity of the alcoholic liver injury progression. This will be beneficial to understand the alcoholic liver injury mechanism and identify new potential therapeutic targets.

In this study, 12 major liver cell types were identified and the changes in gene expression in those cell types were investigated during the progression of alcoholic liver injury. Hepatocytes, endothelial cells and Kupffer cells showed more abnormal DEGs than other types of cells (Figure 3B,D, Appendix A). We found that fatty acid synthesis and coagulation genes of alcoholic liver injury mice were significantly upregulated in hepatocytes (Figure 4C,H), which was consistent with the study of Michael Schonfeld [33]. *Cpt1a* is a rate-limiting enzyme of fatty acid β-oxidation (FAO); the change of its expression or activity will affect liver fat accumulation, and alcohol can reduce its activity and expression [34,35]. In our study, however, *Cpt1a* was upregulated in hepatocytes of alcoholic liver injury mice (Figure 4F). *Cyp4a10* and *Cyp4a14* are known to metabolize arachidonic acid and are significantly increased in ALD patients, promotes lipid accumulation and oxidative stress [36]. For this study, *Cyp4a10* and *Cypa14* were mainly expressed in hepatocytes and significantly increased in alcohol-fed mice (Figure 4F), indicating that liver damage caused by *Cyp4a10* and *Cypa14* mainly occurred in hepatocytes. In this study, the complement genes *C1s1*, *C3*, *C4b*, *C4bp*, *Cfh*, *Cfi*, *Cfhr2*, *Hc* and *Serping1* were significantly increased in hepatocytes of alcohol-fed mice (Figure 4H), suggesting that hepatocytes can produce a large number of complements to establish inflammatory response and fight against alcohol-induced damage. In addition, the coagulation-related genes *F12*, *F2*, *F5*, *F7*, *Fga*, *Fgb*, *Fgg*, *Kng1*, *Kng2*, *Plg*, *Serpinc1* and *Vtn* were significantly up-regulated in hepatocytes of alcohol-fed mice (Figure 4H), indicating that alcohol may enhance the coagulation process of mice. Studies have shown that alcohol can prolong the prothrombin time in male mice [33].

Ethanol is oxidized to acetaldehyde through hepatic alcohol dehydrogenase (ADH) and the microsomal ethanol oxidation system (MEOS), and the oxidation process is dependent on cytochrome P450 2E1 (CYP2E1) [37,38]. MEOS produces reactive oxygen species (ROS) through CYP2E1; this is significantly increased in acute or chronic alcoholic liver injury and facilitates liver injury [39]. Clinical studies have shown that consumption of 40 g ethanol per day for one week in humans leads to increased expression of CYP2E1 [40,41,42]. In addition, recent clinical studies have shown that the CYP2E1 inhibitor chlormethiazole reduces serum AST and ALT levels, and improves steatosis in patients with ALD [43]. In our study, *Cyp2e1* and ADH were mainly expressed in hepatocytes (Appendix A), and alcohol consumption for 3 and 14 days induced the up-regulation of *Cyp2e1* expression (Figure 4C), indicating that short-term alcohol consumption induced liver injury through *Cyp2e1* in hepatocytes. ADH is involved in the oxidative metabolism of ethanol to acetaldehyde, and chronic alcohol consumption leads to a decrease in ADH and further leading to liver injury [44,45]. ADH4 and AHD5 have higher Km (Michaelis—Menten constant) values for alcohol than 30 mM and 100 mM, respectively, while ADH1 has only 0.5–1.0 mM Km for alcohol [41]. In our study, *Adh4* and *Adh5* expression decreased after alcohol treatment, whereas *Aldh1* expression increased at 7 and 14 days of alcohol consumption (Appendix A), suggesting that alcohol decreased the level of ADH in hepatocytes, leading to alcohol accumulation and liver injury.

Considering the activity of TFs, we analyzed the activity of TFs in normal and alcohol groups using SCENIC. Several TFs related to alcoholic liver injury were identified. *Xbp1* is involved in ER stress and is increased in the liver of non-alcoholic steatohepatitis (NASH) patients, increasing fat accumulation and hepatic inflammation [46]. *Xbp1* activity is increased in hepatocytes from mice with alcoholic liver injury (Figure 4J), but the role of alcoholic liver injury in humans requires further investigation. Studies have shown that increased expression of *Stat3* in human alcoholic liver disease patients can further reduce alcoholic liver injury and inflammation [47,48,49]. These studies suggest that increased *Stat3* activity in hepatocytes from alcoholic liver injury plays an essential role in counteracting alcohol-induced liver injury. Strikingly, transcription factors *Spi1*, *Spic*, and *Elf4* in Kupffer cells showed significant increases in activity only at one day of alcohol consumption. *Spi1* is a transcriptional activator that may specifically participate in the differentiation and activation of macrophages or B cells [50,51]. *Spic* inhibits inflammation and participates in macrophage development associated with iron homeostasis [52]. *Elf4* maintains anti-inflammatory genes and inhibits anti-inflammatory gene expression and suppresses inflammatory responses [53]. This implies that Kupffer cells were involved in the hepatic immune reactions in the early stage of alcoholic liver injury (AG1). In addition, transcription factors *Zeb1*, *Sox18*, *Gata4* and *Zmiz1* were activated in AG14 samples, which target genes involved in ameboid-type cell migration, epithelial cell proliferation and response to transforming growth factor beta, implying that these transcription factors are activated in the later stages of alcoholic liver injury (AG14). Whether they are activated in response to longer alcohol stimulation still needs further investigation, however.

In our study, 50,274 liver-single-cell transcriptome data were analyzed. Thanks to scRNA-seq studies, we analyzed and identified cell types that predominantly change during alcoholic liver injury and transcription factors with abnormal activity. This study revealed a small proportion of liver injury in mouse models following short-term ethanol application. However, fewer Kupffer cells and hepatic stellate cells were identified by sequencing the entire cells of the liver, and the key role of these two types of cells remains unclear and warrants further investigation. What is more, the long-term application of ethanol is not mentioned in this study, which will lead to more serious alcoholic liver diseases, such as AFL, AH and HCC.

In summary, this study reveals liver heterogeneity, describes gene expression in liver cell types and provides a comprehensive single-cell transcriptional atlas in alcoholic liver injury. The results showed that continuous alcohol consumption mainly affected gene expression in hepatocytes, endothelial cells and Kupffer cells in mouse liver. In addition, we revealed the important transcription factors for alcoholic liver injury development. These findings help to understand the key molecular mechanisms of alcoholic liver injury pathogenesis and progression and also provide some directions for its prevention and treatment.

## 4. Materials and Methods

### 4.1. Experimental Animal Models

Adult male C57BL/6 mice (23–25 g) were purchased from Hunan Slaughter Jindo Laboratory Animal Co Ltd. (license: SCXK (Xiang) 2019-0004). The mice were fed in an SPF (specific pathogen-free) class environment and under 12 h of light per day in a temperature-controlled environment (22 ± 1 °C, 60–70% humidity). Animals had free access to drink and food during the feeding period. We established alcoholic liver injury model by intragastric administration of 53% alcohol at different times. The experimental alcoholic liver injury model was established by comprehensive reference to other relevant studies and modification [14,16]. In this study, the alcohol used in the model group was 53% alcohol (prepared with absolute ethanol) at a dose volume of 10 mL/kg (equivalent to 5.3 g/kg alcohol), which is the concentration of commonly used commercial liquor, and We established a mouse model of alcoholic liver injury by gavage. The control group was given the same volume of distilled water (10 mL/kg) by continuous intragastric administration. In the study, mice were randomly grouped after one week of acclimatization feeding, with 15 mice in each group as follows: blank control group (BG, continuous gavage of water for 14 days), 1-day alcohol model group (AG1, gavage with water for 13 days and 53% alcohol for 1 day), 3 days alcohol model group (AG3, gavage with water for 11 days and 53% alcohol for 3 days), 7 days alcohol model group (AG7, gavage with water for 7 days and 53% alcohol for 7 days), 14 days alcohol model group (AG14, continuous gavage of 53% alcohol for 14 days). At the end time point of each group, mice were anesthetized with 20% urethane (0.075 mg/g) and blood was taken from the eyes. Blood was centrifuged at 4500× *g* rpm for 15 min at 4 °C to obtain serum after incubating at room temperature for 30 min. About 100 mg of the liver was fixed in 10% formaldehyde solution. All experiments in this study were approved by the Animal Experiment Ethics Committee of Zunyi Medical University.

### 4.2. Serum Biochemical Indicators Assay

The mouse serum total cholesterol (TC), triglyceride (TG), low-density lipoprotein cholesterol (LDL-C), high-density lipoprotein cholesterol (HDL-C), aspartate transaminase (AST), and alanine transaminase (ALT) reagent kits were purchased from Nanjing Jiancheng Institute of Biological Engineering. Measurement of serum indicators according to the manufacturing instructions of the kits.

### 4.3. Histopathological Evaluation

HE staining (hematoxylin and eosin staining) was used to analyze liver tissue for histopathology. The fixed liver tissues were dehydrated in alcohol and xylene sequentially. The dehydrated liver samples were embedded in paraffin, sectioned in the sagittal plane in 6–8 µm sections, and stained with hematoxylin and eosin (H&E). The stained sections were sequentially dehydrated in 80%, 90%, and 100% alcohol, and the coverslips were sealed and observed under the microscope (Olympus BX43, Tokyo, Japan).

### 4.4. Preparation of Single-Cell Suspensions from Mouse Liver

At the end time point of each group, three mice in each group were randomly selected. Mice were anesthetized using urethane and perfused with a two-step liver perfusion protocol [54,55]. After perfusion, the cells were pushed through a sterile 40 µm filter, and separated into individual cells. Hepatocytes were spun and collected at 50 g for 1 min at 4 °C. The suspension was centrifuged at 350× *g* for 5 min at 4 °C to collect NPCs. Hepatocytes and NPCs were resuspended using DMEM complete medium. Hepatocytes were added to these NPCs to give a final hepatocyte concentration of approximately 10% of the total cell number. The viability of the mixed cells should be higher than 85%, and the cell concentration was adjusted to 1000 cells/μL [56]. Cells not immediately sequenced were stored frozen at −80 °C.

### 4.5. Single-Cell Library Construction and Sequencing

The single-cell RNA-seq libraries were prepared with Chromium Next GEM Single Cell 3′ Reagent Kits v3. Briefly, the prepared single-cell suspension was combined with the barcoded mRNA capture beads, droplet generation oil and the mixture of enzymes. Then it was encapsulated in the “double cross” droplets of microfluid to form gel bead-in-Emulsions (GEMs). Cell lysis and reverse transcription reactions were performed in GEM. The GEMs were broken up and collected by the bead filter, and PCR amplification was performed using cDNA as the template. The quality of amplification products was checked (the size of amplification fragments and the output of amplification products). After the amplification products were qualified, the Chromium 3’v3 kit (10× Genomics) was used to construct the sequencing library. Finally, after the library was completed, we checked the database, and used Illumina HiSeq sequencing platform for sequencing to obtain the sequencing data and subsequent data analysis.

### 4.6. scRNA-Seq Data Processing

The raw data were compared to the mouse reference genome (mm10-3.0.0) using Cell Ranger (v 3.1.0) provided by 10×Genomics. Next, cellranger was used to generate the UMI matrix. The Seurat (https://github.com/satijalab/Seurat/ (accessed on 14 March 2022)) R package was used for quality control, dimensionality reduction, and clustering analyses. The subset function screened and filtered low-quality and abnormally expressed cells. First, cells expressing less than 200 and over 6000 genes were excluded. Second, dead cells identified as cells with more than 25% reads coming from mitochondrial genes were excluded. Third, erythrocytes identified as cells with more than 1% of reads coming from mitochondrial genes were removed. Fourth, cells with a UMI number less than 500 and greater than 40,000 per cell were excluded from the analysis. After data cleaning, the resulting filtered UMI matrices were transformed into Seurat objects with the function CreateSeuratObject with the UMI matrix as counts, min.cells = 3, min.features = 300. Cell cycle was assessed with the R function CellCycleScoring, with s.features and g2m.features provided by Seurat in the R object cc.genes after being transformed into mice genes. The data were normalized, the highly variable genes were identified and scaled with the function SCTransform with parameters vars.to.regress = “percent.mt”, considering all the cells. All five Seurat objects were integrated using the CCA algorithm used by FindIntegrationAnchors and IntegrateData functions in Seurat. We performed a Principal Component Analysis (PCA) of the Seurat object with the R function RunPCA for all the cells. Euclidean distance K-nearest neighbors (KNN) were constructed using the FindNeighbor function to refine the boundary weights between cells and delineate cells with similar gene expression patterns, and then FindClusters function (resolution = 0.8) was used to identify cell subpopulations. An unsupervised clustering with a t-distributed stochastic neighbor embedding (t-SNE) analysis was performed on the transcriptomes using the R function RunTSNE with parameters dims = 1:20 (All cells). The cell type-specific genes of each cluster were identified with the FindAllMarkers function with parameters min.pct = 0.25, logfc.threshold = 0.25, test.use = ‘MAST’. Top-ranked genes were ordered by fold change under a threshold of expression of at least 25% of cells, fold change greater than 1.5-fold and adjusted *p* value < 0.01. Enrichment analysis was performed using the R packages ReactomePA on top-ranked genes [57]. Finally, according to cell type-specific genes, combined with marker genes from literature reports, SingleR and CellMarker databases, cell type identification was carried out to annotate cell subsets.

### 4.7. Differential Expression Analysis

DEGs analysis between alcoholic liver injury mice and control mice was performed using the function FindMarker in Seuart, using a MAST test [58]. Genes with a *p* value less than 0.05, expressed in at least 25% of cells were considered to be differentially expressed. Functional enrichment analysis of all DEGs was performed using the enrichGO function in the R package clusterProfiler [59].

### 4.8. SCENIC Analysis

We applied single-cell regulatory network inference and clustering (SCENIC) analysis to identify transcription factors (TFs) for some cell types (hepatocyte, endothelial cells and Kupffer cells) [60]. For each cell type, we performed SCENIC analysis for five groups, respectively. The regulon s and TF activity for each cell were calculated with motif collection version mc9nr (10 kb up and down, and 500 bp up and 100 bp down) from the cisTarget (https://resources.aertslab.org/cistarget/ (accessed on 19 October 2022)). Gene regulation of cells was constructed using the R package GENI, RcisTarget and AUCell. When focusing on the activity of TFs, transcription factors that activated in at least 70% of cells in at least one group and *p* value < 0.01 compared to the BG group were analyzed.

### 4.9. Quantification and Statistical Analysis

All quantification data were expressed as mean ± SEM. Statistical analyses were performed utilizing R software (version 4.1.1). All parameters were analyzed by Student’s *t*-test and one-way ANOVA with R; a threshold of *p* < 0.05 was considered statistically significant.

## Figures and Tables

**Figure 1 ijms-24-04344-f001:**
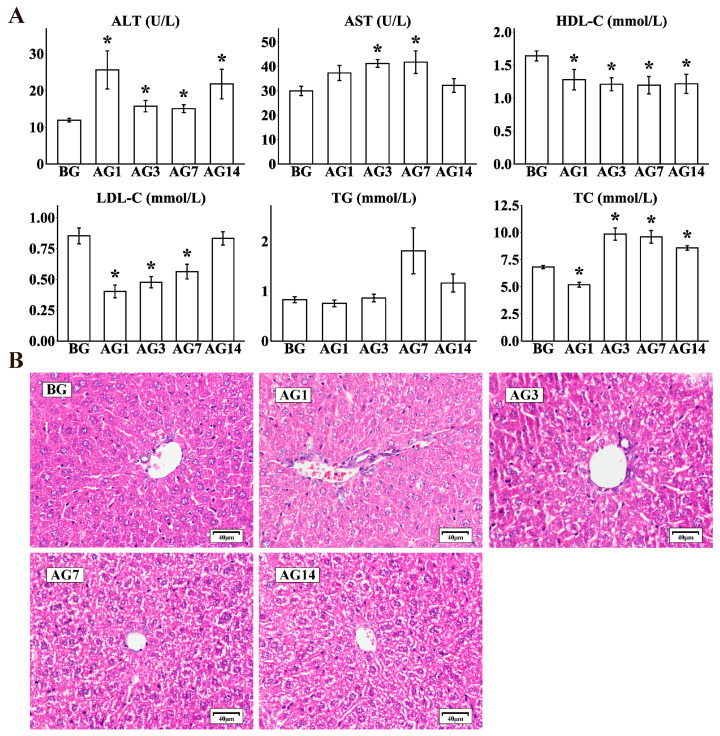
Serum index and HE staining of liver in mice with alcoholic liver injury. (**A**). Serum total cholesterol (TC), Triglyceride (TG), Low-Density Lipoprotein Cholesterol (LDL-C), High-density lipoprotein cholesterol (HDL-C), Alanine transaminase (ALT) and Serum aspartate aminotransferase (AST) were detected using a microplate technique (n = 10 per group), * *p* < 0.05 vs. BG. (**B**). Liver morphology was detected by hematoxylin-eosin (H&E). Scale bar = 40 μm.

**Figure 2 ijms-24-04344-f002:**
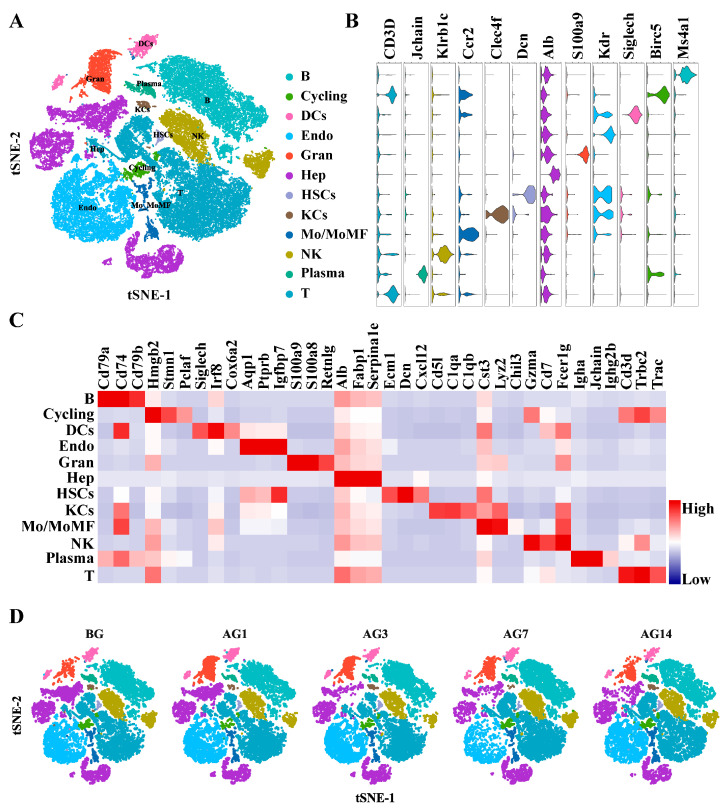
Clustering and annotation of liver transcriptome in mice with alcoholic liver injury. (**A**) A total of 50,274 mouse liver single cell transcriptome and cell-type annotation of each subpopulation based on the differential expression of liver cell type-specific genes. (**B**) Expression of representative enriched genes for each cell type. Gene expression violin plots are shown in log-scale UMI. Colors correspond to cell types. (**C**) Heatmap displaying the expression level of established liver cell type-specific top 3 markers in each cell type. (**D**) The t-SNE plot shows the different distribution of 12 clusters in BG, AG2, AG3, AG7 and AG14 mice.

**Figure 3 ijms-24-04344-f003:**
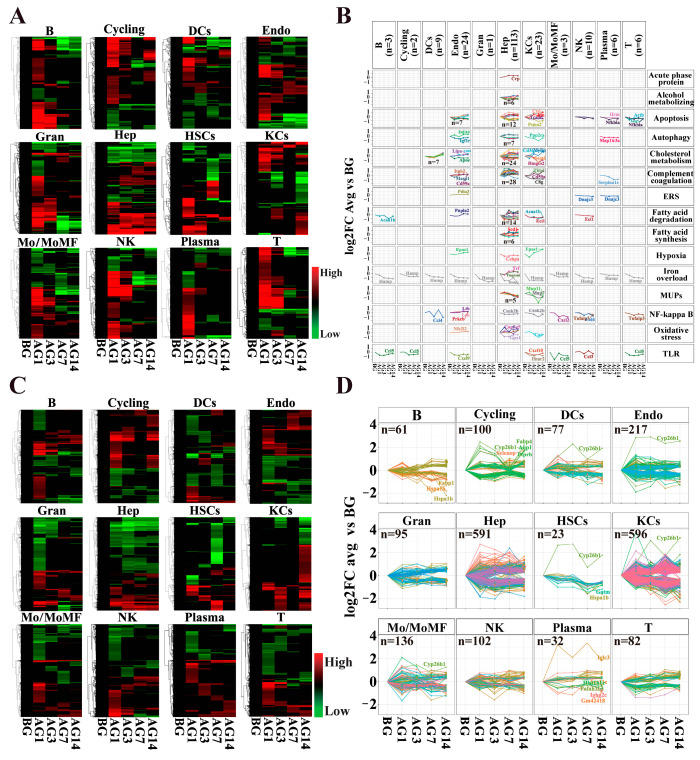
The alcohol-induced liver injury mainly occurred in hepatocytes, endothelial cells and Kupffer cells. (**A**) Heatmap of gene expression of ALD-related genes in different liver cell types. (**B**) Expression of ALD-related genes in liver cells. Colors represent different genes. Dots indicate statistically significant differences compared with BG group (*p* < 0.01). The value used in the figure is the logarithm based on 2 of the ratios of the average expression of each cell to the average expression of the BG group. (**C**) Heatmap of DEGs expression in different liver cell types. (**D**) Expression of DEGs in liver cells. Different colors represent different genes and the dot indicates that there is a significant difference with the BG Group (*p* < 0.01). The values used in the figure are based on the logarithm of 2 of the mean expression ratios of each cell type to the BG group.

**Figure 4 ijms-24-04344-f004:**
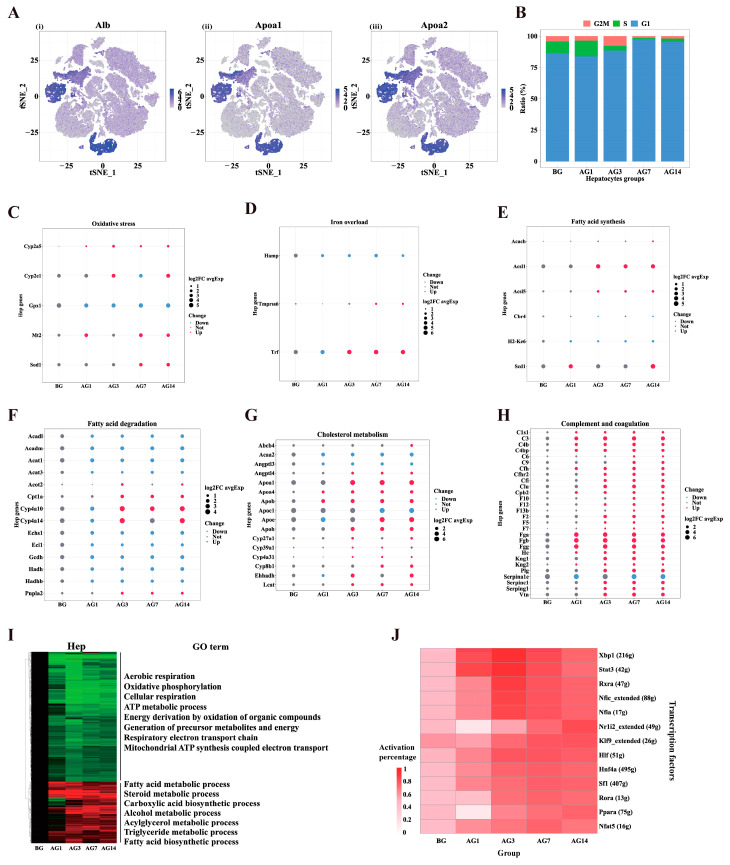
Effect of alcohol on gene expression in mouse hepatocytes. (**A**) Gene expression of hepatocyte markers, (i) *Alb*, (ii) *Apoa1*, (iii) *Apoa2*. Color bars indicate the expression level in log-scale UMI. (**B**) Distribution of cell cycle phases (G2/M, S, G1) in hepatocytes of different groups. (**C**–**H**) Dot plots showed the relative expression changes of genes related to oxidative stress, iron overload, fatty acid synthesis, fatty acid degradation, cholesterol metabolism, complement and blood coagulation in different groups of hepatocytes. The size represents the average Log2FC value compared to the BG group. (**I**) Differential expressed genes and GO terms between AG1, AG3, AG7and AG14 compared to BG in hepatocytes. (**J**) Heatmap of the percentage of cells with the regulon active in different groups of hepatocytes.

**Figure 5 ijms-24-04344-f005:**
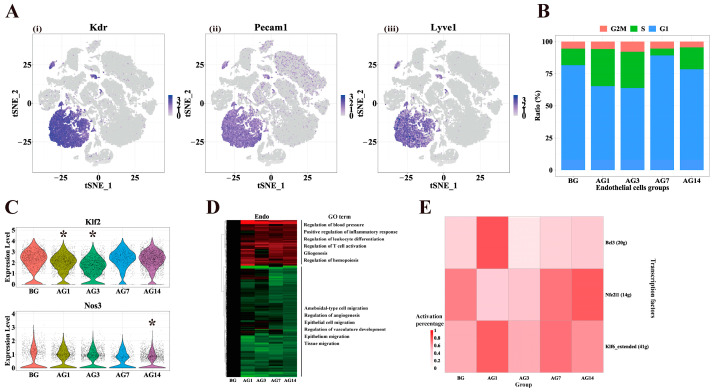
Aberrant gene expression and aberrant activity transcription factors in endothelial cells. (**A**) Gene expression of endothelial cells markers, (i) *Kdr*, (ii) *Pecam1* and (iii) *lyvel*. Color bars indicate the expression level in log-scale UMI. (**B**) Distribution of cell cycle phases (G2/M, S, G1) in endothelial cells of different groups. (**C**) Violin plots showing expression levels of NO metabolism genes in endothelial cells of control and alcoholic liver injury mice, * *p* < 0.01 vs. BG. (**D**) Differential expressed genes and gene functions between AG1, AG3, AG7 and AG14 compared to BG in endothelial cells. (**E**) Heatmap of the percentage of cells with the regulon active in different groups of endothelial cells.

**Figure 6 ijms-24-04344-f006:**
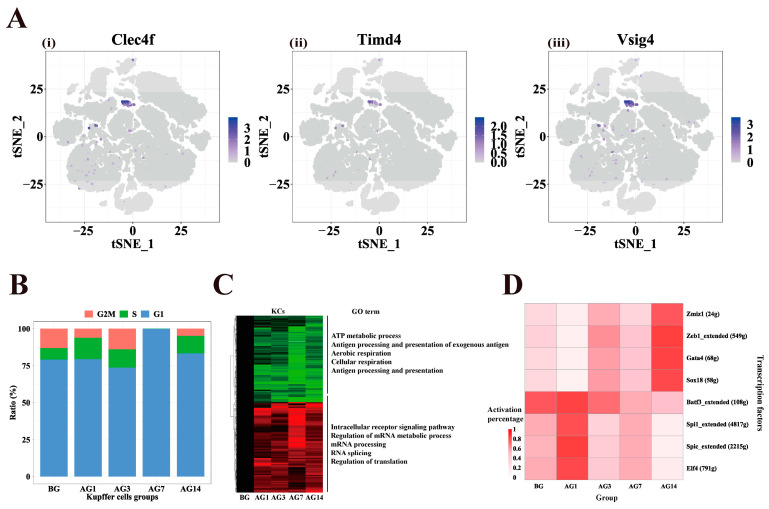
Abnormally expressed genes and transcription factors in Kupffer cells. (**A**) Gene expression of Kupffer cells markers (i) *Clec4f*, (ii) *Timd4* and (iii) *Vsig4*. Color bars indicate the expression level in log-scale UMI. (**B**) Distribution of cell cycle phases (G2/M, S, G1) in Kupffer cells of different groups. (**C**) Differentially expressed genes and gene functions between AG1, AG3, AG7and AG14 compared to BG in Kupffer cells. (**D**) Heatmap of the percentage of cells with active regulators in different groups of Kupffer cells.

## Data Availability

The data generated and analyzed in this study are available from the corresponding author on request.

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
