# Peer review of "Single-Cell RNA Transcriptome Profiling of Liver Cells of Short-Term Alcoholic Liver Injury in Mice"

_ijms, 2023, doi:10.3390/ijms24054344_

Round 1

Reviewer 1 Report

The investigators applied well-established ssRNAseq methodology to mouse models of alcoholic liver disease. The novelty of this work derives largely from the application of ssRNAseq technology to ALD models. 

I found the design and execution of experiment to be rigorous and thorough. I was particularly impressed with the use of multiple alcohol treatments, and the use of 15 mice per group, and I found the handling of statistics was adequate and it was described in sufficient detail. Although it should be noted that only male mice were used. 

These data will be of great interest to investigators in the ALD field, thus it is imperative that the authors provide easy access to the complete dataset following publication. 

My only criticism of the manuscript is that I would like to have seen more in-depth discussion and comparison to previous liver ssRNAseq datasets. I think one of the more novel findings was that hepatocytes, endothelial cells, and Kupffer cells responded most to alcohol. However, based on the discussion it was difficult to assess the novelty of the findings regarding affected gene ontology pathways, which are well documented in the literature. For example the impact on lipid metabolism, oxidative stress, hypoxia, complementation and anticoagulation, and energy metabolism on hepatocytes are well described in the literature, but perhaps there are more detailed insights to be gleaned from such a rich dataset, when contrasted with other ssRNAseq datasets.

I am especially interested in what appears to be three distinct populations of hepatocytes (Fig. 2A). Previous work, such as Halpern et al 2017, were able to resolve and identify zone specific markers within the hepatocyte population. Are the hepatocyte clusters shown in Figure 2 periportal versus pericentral hepatocyte populations? I bring this point to the authors’ attention because I feel that it will be of interest to many readers, but in general more comparison to existing liver ssRNAseq datasets would be appreciated and may extract additional insights from this highly complex dataset.

Reviewer 2 Report

This report by Cao et al. describes the gene expression changes in specific liver cell types (via single cell RNA seq) following adminstration of alchohol to mice.  Although the data would be of general interest to investigators studying ALD, the report appears to be somewhat lacking in any real mechanistic insights into this disease.  Specific comments are below:

1.     The authors are to be commended for their efforts in putting together this manuscript, however, the english grammar needs to be improved.  I would strongly suggest enlisting a professional editing service or from someone fluent in English. 

2.     The last paragraph of the introduction should only state what the authors did and not have any results presented.

3.     Need a better interpretation of the H&E sections, using standard terms (e.g. balooning, etc.)

4.     It is unusual to present ALT and AST data as fold-change.  Typically, the values in U/L are shown.

5.     In mice, ALT and AST values generally follow the same trend, but with AST values being larger than ALT.  Can the authors explain why in some time points there is a change in ALT that is not seen in AST?

6.     Their mouse model of ALD is not clear.  They use the intragastric, and occasionally “continuous intragastric”.  Do they perform a gavage or did they use the Tsukumoto model?  If they gavaged, how much did they provide to the mice (grams of ethanol per kilogram of body weight).  A 53% ethanol solution seems a bit high.  Typically solutions no greater than about 35% are used owing to the discomfort (i.e. “burning”) that high concentrations of alcohol elicits.  Much more detail is needed on their model. 

7.     Did the authors independently validate any of the gene changes observed, either through qPCR of isolated cells (hepatocytes, for example) or via IHC or Western analyis?  Given the short duration of the model, this could have easily been accomplished.

8.     Sections 4.4 and 4.5 have the same subheading.  Is this the way the authors intended or is it a mistake?

9.     It seems that by using such a sophisticated approach one would identify unique mechanisms driving the onset or progression of alcoholic liver disease.  Beyond a cataloguing of genes in different cell types, there isn’t any real breakthrough in our understanding of ALD.

Reviewer 3 Report

This manuscript reported the single-cell RNA transcriptome profiling of liver cells of short-term alcoholic liver injure in mice. My major concern is the author based on what are the justification(s) to determine the days of treatment and dosages of alcohol used in this study. How these findings related to alcohol consumption induced liver injury in humans for particular alcohol related disease(s) or pathological condition(s)? Worst still, as shown in Figure 1A, the short-term alcohol consumption induced liver injury in mice was very mild and varies between time points (lack of significant trends for most parameters especially for ALT and AST). In addition, the authors only superficially describe the changes of gene expression of the different kind of cells without any validation. Whether the changes are the cause, consequence or just meaningless correlation to the short-term alcohol consumption induced liver injury remain to be explored.

Round 2

Reviewer 2 Report

The authors have satisfactorily addressed my concerns in their revised manuscript. 

Author Response

Thank you for your nice comments on our manuscript. We appreciate your warm work earnestly and approval of our manuscript. 

Reviewer 3 Report

For point 1, the authors failed to convince me that their acute alcohol treatment mouse model mimics the situation if humans, as no related publications were cited to support their claims.

For point 2, the authors are not addressing my question. They just rephrase my comment and without explain how these findings are related to alcohol consumption induced liver injury in humans. 

For point 3, Will tolerance be developed within two weeks?

For point 4, only one is validated that is not acceptable.
